# A 16-week randomized controlled trial of a fish oil and whey protein-derived supplement to improve physical performance in older adults losing autonomy—A pilot study

Anne-Julie Tessier[1,2], Julia Lévy-Ndejuru[1], Audrey Moyen[1], Marissa Lawson[1], Marie Lamarche[2], Joseé A. Morais[1,2,3], Amritpal Bhullar[4], Francis Andriamampionona[5], Vera C. Mazurak[4], Stéphanie Chevalier [1,2,4] *

1 School of Human Nutrition, McGill University, Ste-Anne-de-Bellevue, Quebec, Canada, 2 Research Institute of the McGill University Health Centre, Montreal, Quebec, Canada, 3 Division of Geriatric Medicine, McGill University, Montreal, Quebec, Canada, 4 Department of Agricultural, Food and Nutritional Science, University of Alberta, Edmonton, Alberta, Canada, 5 Institut universitaire de gériatrie de Montréal, Montréal, Quebec, Canada

* stephanie.chevalier@mcgill.ca

**Data Availability Statement:** Due to the small sample size of our data set, and to respect the

## Abstract

### Background

Low functional capacity may lead to the loss of independence and institutionalization of older adults. A nutritional intervention within a rehabilitation program may attenuate loss of muscle function in this understudied population.

### Objective

This pilot study assessed the feasibility for a larger RCT of a nutritional supplementation in older adults referred to an outpatient assessment and rehabilitation program.

### Methods

Participants were randomized to receive a supplement (EXP: 2g fish oil with 1500 IU vitamin D3 1x/d + 20-30g whey protein powder with 3g leucine 2x/d) or isocaloric placebo (CTR: corn oil + maltodextrin powder) for 16 weeks. Handgrip and knee extension strength (using dynamometry), physical performance tests and plasma phospholipid *n*-3 fatty acids (using GCMS) were evaluated at weeks 0, 8 and 16; and lean soft tissue mass (using DXA), at weeks 0 and 16.

### Results

Over 2 years, 244 patients were screened, 46 were eligible (18.9%), 20 were randomized, 10 completed the study (6 CTR, 4 EXP). Median age was 87 y (77–94 y; 75% women) and gait speed was 0.69 m/s; 55% had low strength, and all performed under 420m on the 6-minute walk test, at baseline. Overall self-reported compliance to powder and oil was high (96% and 85%) but declined at 16 weeks for fish oil (55%). The EXP median protein intake

confidentiality of our participants as stated in the informed consent form they have signed, the data underlying this study cannot be made publicly available. Data access requests may instead be sent to the Research Ethics Board (CT2) of the Research Institute of the McGill University Health Centre, which approved this study: Sheldon Levy, REB Coordination Email: Sheldon.levy@muhc.mcgill.ca Telephone: (514) 934-1934 ext. 36077.

**Funding:** This work was supported by the Helen McCall Hutchison Award in Geriatric Medicine awarded to S.C. and Réseau québécois de la recherche sur le vieillissement of the Fonds de Recherche du Québec – Santé (FRQS). http://www.rqrv.com/en/index.php A.-J.T. was awarded a PhD Graduate Excellence Fellowship by McGill University, a PhD fellowship from the Research Institute of the McGill University Health Centre (RI-MUHC; https://rimuhc.ca) and a Doctoral training fellowship for Applicants with a Professional Degree from the FRQS; http://www.frqs.gouv.qc.ca. J.L.N was awarded an Undergraduate Summer Studentship from the Canadian Institutes of Health Research (CIHR; https://cihr-irsc.gc.ca). The funders had no role in study design, data collection and analysis, decision to publish, or preparation of the manuscript.

**Competing interests:** The authors have declared that no competing interests exist.

surpassed the target 1.2–1.5 g/kg/d, without altering usual diet. Proportions of plasma phospholipid EPA and DHA increased significantly 3- and 1.5-fold respectively, at week 8 in EXP, with no change in CTR. Participants were able to complete most assessments with sustained guidance.

## Conclusion

Because of low eligibility, the pilot study was interrupted and deemed non-feasible; adherence to rigorous study assessments and to supplements was adequate except for long-term fish oil. The non-amended protocol may be applied to populations with greater functional capacity.

## Trial registration

ClinicalTrials.gov NCT04454359.

## Introduction

Older adults represent the fastest-growing age group worldwide, and increasing life expectancy contributes to a burgeoning population of very old adults [1]. Aging is associated with sarcopenia and functional decline; from the age of 50 y, 0.5–1% of muscle mass and 2–3% function are lost yearly [2]. Especially in later life, this may lead to frailty, loss of independence, hospitalization and mortality [3]. With multifactorial causes, i.e. physical, cognitive, mental and social, the loss of independence is commonly defined by difficulties in performing activities of daily living [4]. It is often the leading reason for institutionalization.

To foster *aging well at home and in the community*, there is urgent need to identify and offer strategies such as nutritional interventions for improving functional capacity and maintenance of an independent life. Indeed, malnutrition is highly prevalent in the elderly population with up to ~30% being observed in rehabilitation settings [5]. Poor nutritional status has been associated with impaired physical function capacity and frailty [6]. Resistance-type training without [7] and with [8] protein supplementation was shown to promote gains in muscle mass and strength in younger and older adults. But, nutrition alone may represent an addressable risk factor and a suitable approach in older adults who cannot or do not want to engage in such exercises [9]. The effect of protein supplementation on the muscular system may largely depend on protein quality, dose and timing of ingestion [9], and presence of anabolic resistance. Other nutrients are of interest for their potential role on muscle. Indeed, high-dose vitamin D supplements (800–1000 IU) were shown to have a favorable effect on balance, strength and physical performance in elderly populations. Improvements in physical performance and lean mass were also reported following supplementations with *n*-3 fatty acids in healthy older adults [9].

Few studies have tested the effect of combined-nutrient supplements and found modest but significant effects on total lean mass, strength and physical performance in sarcopenic older adults [10, 11] or healthy men [12]. A nutrient-dense supplement may represent a practical approach for a short-term intervention as part of a geriatric rehabilitation program and may have additive beneficial effects on muscle parameters and physical function in very old adults with low functional capacity referred for assessment of loss of independence. This population is heterogenous, presenting with either sarcopenia, dynapenia, frailty, mobility limitations and

low functional capacity, hence is challenging to study. Therefore, this pilot study aimed to assess whether an RCT of a combined nutritional supplementation is feasible with regards to recruitment, compliance and completion of assessments. Our secondary outcome was to characterize the nutritional and functional status of this specific population of older adults.

## Methods

### Participants

Participants were free-living older adults referred to an outpatient rehabilitation program that includes a comprehensive geriatric assessment to evaluate their level of independence and aims to improve it, to delay institutionalization. Participants were referred to the program either by the local community health center responsible for delivering home care assistance, or upon returning home after hospitalisation (commonly due to falls). The study was conducted at the Geriatric Day Hospitals (GDHs) of the McGill University Health Centre (MUHC) Montreal General Hospital and of the *Institut universitaire de gériatrie de Montréal*, QC. All new patients admitted to the GDHs were screened and approached if they met eligibility criteria. Eligible patients were able to read and speak English or French and had a MMSE >22/30. Patients with a BMI >35 kg/m$^2$, presence of kidney (eGFR <30 mL/min/1.73 m$^2$), liver or heart failure, stroke in the last 6 months, Parkinson's disease, severe neuropathy, active malignancies, acute inflammation (CRP >10 mg/L), diagnostic of hyperparathyroidism, recent acute weight loss (>10% in 3 months, unless stabilized), allergy to milk and/or fish, long-term use of corticosteroids or anti-neoplastic medication were excluded. Eligible patients were invited to review study procedures and sign the consent form. For the duration of the study, vitamin D supplementation dose was changed or kept at 400 IU/d for all participants one week prior to the study and those taking n-3 PUFA supplements were asked to cease it. This study was approved by the McGill University Health Centre Ethics Board (REB 15-633-MUHC), was registered on ClinicalTrials.gov (NCT04454359) and followed the Consolidated Standards of Reporting Trials (CONSORT checklist, S4 Table). This trial was registered after participant recruitment began due to the pilot nature of the study. The authors confirm that there are no ongoing or related trials for this intervention which required further registration.

This pilot study was not intended to evaluate the efficacy of the intervention and is not powered to detect significant differences in outcomes. Sample sizes between 10–40 participants per group have been recommended for pilot trials [13]. We aimed to recruit 40 within 1 year based on the expected rate of GDH patient annual turnover to provide confidence intervals (CI) to establish sample size for the large RCT.

### Study design

Participants attended the rehabilitation program for 8 weeks on average; they would come at the GDH for half a day, twice per week, to meet with health professionals for general health and functional status assessment. Depending on individual needs, they performed few exercises determined by the physiotherapist during their consultation and were recommended individualized exercises to perform at home. Participants were randomly assigned to receive either the multi-nutrient supplement (EXP) or the placebo (CTR) for 16 weeks, combined with the rehabilitation program for the first 8 weeks and after program completion for 8 weeks. Simple randomization coded for the EXP and CTR group (1:1) was generated by a staff member external to the study using www.randomization.com. The study coordinator, the investigators and participants were all blinded to group allocation.

Visits for assessments of participants before the study, at weeks 8 and 16 were performed at the Research Institute of the MUHC during which physical tests, body composition assessment

and questionnaires were completed. Overnight fasting blood was drawn during study visits or at home by a registered nurse or physician at all study time points. Adapted transport, or taxi was planned, and expenses covered. Supplements were planned to be delivered at the GDHs fortnightly by a research staff who would also collect used containers from participants. Participants were called by phone at least weekly for monitoring.

## Intervention

The EXP group supplement consisted of 25 g, 30 g or 35 g flavored whey protein isolate (New Zealand Whey Protein Isolate, The Protein Company ATW Inc, QC, Canada) scaled to the participant's weight (65 kg, 65–75 kg, and 75 kg respectively) with a fixed 3 g of added leucine. The CTR group supplement was an isocaloric placebo of 25 g, 30 g or 35 g maltodextrin (S3 Table). Both products were vanilla-flavoured powders and were provided as individual pre-weighted doses in opaque containers. Participants were instructed to mix the powder with 125 mL water before ingestion and to consume it twice daily, once before breakfast for a more even mealtime distribution of protein intake [14] and the other before bedtime to potentially reduce overnight protein catabolism [15]. The total amount of protein supplemented aimed to increase participants protein intake to 1.2–1.5 g/kg/d [16]. The n-3 PUFA with vitamin D supplement was provided as a fruit-flavored oil (NutraSea Liquid®, Ascenta, NS, Canada). Participants received a dosing cup, pre-marked to measure 7.5 mL, providing 1500 IU vitamin $D_3$ and 1125 mg eicosapentaenoic acid (EPA)+750 mg docosahexaenoic acid (DHA) as triglycerides and were instructed to consume this amount once daily. These doses were selected based on prior work demonstrating maintenance or gain of muscle mass in cancer patients undergoing chemotherapy with ingestion of these doses over 10 weeks [17]. The vitamin D dose was chosen to be higher than an effective dose of 800 IU/d associated with decreased risks of falls [18] and is commensurate with prescribed vitamin D supplements of 10,000 IU per week, to adults of 65 years and older in the province of Quebec. An isocaloric corn oil placebo was provided to the CTR group (S3 Table). Both the treatment and placebo oils were provided in amber bottles.

## Outcomes measures

The main outcome measures of this pilot study were feasibility as evaluated by the recruitment, eligibility, consent, attrition rates and completion of assessments by the participants, and acceptability as measured by the self-reported and objective compliance rates. Specific thresholds were established for each criterion prior to commencing the study: recruitment ≥50%, eligibility ≥30%, consent ≥50%, and attrition rates <10%, completion of study outcome assessments ≥80% and compliance ≥80%.

A logbook was provided to participants to report the amount of powder and oil supplements taken daily. All powder and oil containers were collected back at the GDH or at the participants' home for convenience, at the time of biweekly supplement provision. Empty and full containers were compiled and used to define self-reported adherence. N-3 PUFA proportions in the plasma phospholipid fraction was used as an objective measure of fish oil compliance [19].

Blood was collected in SST and EDTA-K2 vacutainers, kept on ice, and centrifuged within 1 hour of collection. Serum and plasma were aliquoted and kept at -80°C until analysis. Serum albumin, pre-albumin, creatinine, CRP were measured by the MUHC Central Biochemistry Laboratory using standard methods and serum 25(OH)D, by chemiluminescence immunoassay (CLIA; Beckman Dxl 800 Series, USA). Plasma glucose was measured by the glucose

oxidase method (GM9 Glucose Analyzer, Analox™) and serum insulin by ELISA (Mercodia™). Insulin-like growth factor I was measured by ELISA (R&D Systems™).

Plasma phospholipid fatty acid profile and quantitative analyses were performed as described in [20]. Briefly, following a modified Folch method, plasma lipids were extracted and the phospholipid fraction was separated by thin-layer chromatography. The internal standard C17:0 was added for quantification followed by methylation and analysed by gas chromatography-flame ionization (Varian Instruments, Canada). Mean fatty acid amounts were calculated from duplicates; $\leq$ 5% variation between duplicates was considered acceptable.

Total lean soft tissue (lean mass) and fat mass were measured by dual-energy X-ray absorptiometry (DXA; GE Lunar iDXA, GE Healthcare). Appendicular lean mass (ALM) was calculated as the sum of the four limbs lean mass and ALM index, as ALM over height squared (kg/$m^2$). Calibration verification throughout the reportable range was completed on the morning of scheduled scans by a trained technician. All images were inspected for proper delimitation of limbs and re-analysed as needed. DXA scans were performed at baseline and week 16.

Participants were classified as sarcopenic if their ALM index fell <7.31 kg/$m^2$ for men and <5.43 kg/$m^2$ for women, as per Canadian cut-points for sarcopenia [21]. Body weight was measured using a digital scale (Scale-Tronix USA; nearest 0.1 kg) in light clothing and at the same time of day. Standing height was measured to the nearest 0.1 cm using a stadiometer as per standard procedures. BMI and waist circumference were measured according to standard procedures at all 3 study time points.

Maximal handgrip strength was evaluated by hand-held dynamometry (Jamar hydraulic, USA) [22]. Participants were seated in a chair without armrests, elbow flexed at a 90-degree angle and were instructed to squeeze the device as hard as they could for 5 seconds; 3 measurements were performed with each hand, alternating sides. The highest measure was used for analysis and the maximal results from the same hand were compared over time. Participants were classified as having low handgrip strength according to the sex-specific Canadian cut-points for dynapenia, <33 kg for men and <20 kg for women [21]. Maximal leg strength was measured using isometric knee extension test (Biodex System 4 Pro, Biodex Medical Systems Inc.) [23]. Three 5-second contractions were performed with leg placed at a 60-degree angle, alternating with a 5-second rest. The peak torque was recorded in Newton-meters (Nm).

The timed-up and go (TUG) test was used to assess mobility [24]. Participants were instructed to stand up from an armchair, walk a 3-meter distance, turn around, walk back to the chair and sit. The test was repeated twice and the average time in seconds used in analyses. The 6-minute walk test (6MWT) [25] was performed in a 30-meter-labeled corridor. Participants walked back and forth this distance within a timed 6-minutes. The distance covered was recorded to the nearest 0.5 meter. The test was stopped if the participant needed to sit. Gait speed was measured within the 6MWT to avoid exhaustion. To eliminate acceleration and deceleration, a tape was placed at 4 and 8 meters from the starting line. The time (seconds) required to cover the 4-meter distance was recorded; gait speed was reported in m/s [26]. The use of usual daily living assistive devices was permitted for the 6MWT/gait speed and TUG. The 30-second chair-stand was used to assess lower limbs strength and balance [27]. With arms crossed on their chest, participants had to rise from a chair to a full standing position and sit back as many times as they could within 30 seconds. The number of complete stands were recorded.

Dietary intake was assessed with three-day food diaries (3DFD) and analysed using the Food Processor software (ESHA®; Canadian Nutrient File v2015). Participants received instructions to record food intake and estimate portion sizes using measuring cups. Diaries were verified by a dietitian.

Daily average step counts were calculated using accelerometers (ActiGraph, GT3x) that participants wore during 4 consecutive days. Participants kept an activity log to verify concordance with the accelerometer results. Nutritional status and symptoms of depression were evaluated through the Mini-Nutritional Assessment-Short Form (MNA-SF) [5] and Geriatric Depression Scale (GDS), respectively. Global cognitive status was assessed using the MMSE and the Montréal Cognitive Assessment (MoCA) [28]. Frailty was determined as per the Fried criteria [29].

## Statistical analysis

For the primary outcomes including recruitment, eligibility, consent and attrition rates, 95% CI for a population proportion were calculated and compared to expected rates. Medians and 95%CI were used to evaluate adherence rates. Medians and ranges were reported for continuous variables and raw counts for nominal data. All statistical analyses were performed in Python 3.0 using Pandas and NumPy libraries.

## Results

Of 244 patients screened, 61 were approached and further assessed for eligibility. Between August 2016 and August 2018, 23 agreed to participate, 20 were randomized and 13 received the allocated group intervention. Ten participants completed the study which ended in September 2018. Fig 1 shows the CONSORT flow diagram of participants as per the extension statement for pilot and feasibility trials [30]. The recruitment rate was 1 participant every two months (95%CI: 0.6, 1.4), 14% of the expected monthly recruitment rate; over 2 years, the eligibility rate was 19% (95%CI: 14, 24), consent rate, 46% (95%CI: 32, 60) and attrition rate, 23% (95% CI: 0, 46). Reasons for non-eligibility were mostly neuropathy, impaired cognitive function, corticosteroids use and early abandon of the rehabilitation program. Of the 23 patients who consented to participate, 2 were excluded shortly after baseline assessments and before starting the supplementation, for high serum CRP level and need for wheelchair; data were included in baseline characteristics.

The population studied (Table 1) had a median age of 87 (77–94) and predominantly comprised women (75.0%). At baseline, 26% had low lean mass, 55% had low handgrip strength, 26% were frail, 85% had mobility limitations requiring a walking aid on a daily basis. All participants had low functional capacity based on the 6-minute walk test, below 415 m [31]. The median 25-hydroxyvitamin D was 74.0 nmol/l (44.0–159.0), above the sufficiency level of 50 nmol/L [32]. Table 2 shows baseline and follow-up characteristics and outcome measures data by group. Based on 95%CI (not shown), there were no differences in baseline characteristics between groups. One participant had a BMI slightly >35 kg/m$^2$ but was included in the study to benefit from the study intervention and increase recruitment.

### Adherence and adverse events

The overall median adherence to the powder and oil supplements as evaluated by leftover count was 95.6% (mean: 87.2%, 95%CI: 74.9, 99.6) and 85.1% (mean: 67.4%, 95%CI: 45.1, 89.7), respectively. When analyzing groups separately, adherence to powder was 99.1% (mean: 96.0%, 95%CI: 90.3, 101.7) in CTR compared to 93.3% (mean: 92.6%, 95%CI: 87.6, 97.7) in EXP and 94.0% (mean: 82.8%, 95%CI: 58.9, 106.8) compared to 70.2% (mean: 68.8%, 95%CI: 40.1, 97.4) to oil; oil supplement intake decreased from 85.7% at week 8 to 54.8% at week 16 in EXP.

Proportions of plasma phospholipid EPA and DHA significantly increased in EXP at week 8 by 3 and 1.5 folds, respectively (Fig 2 and S1 Table). Only the DHA change remained

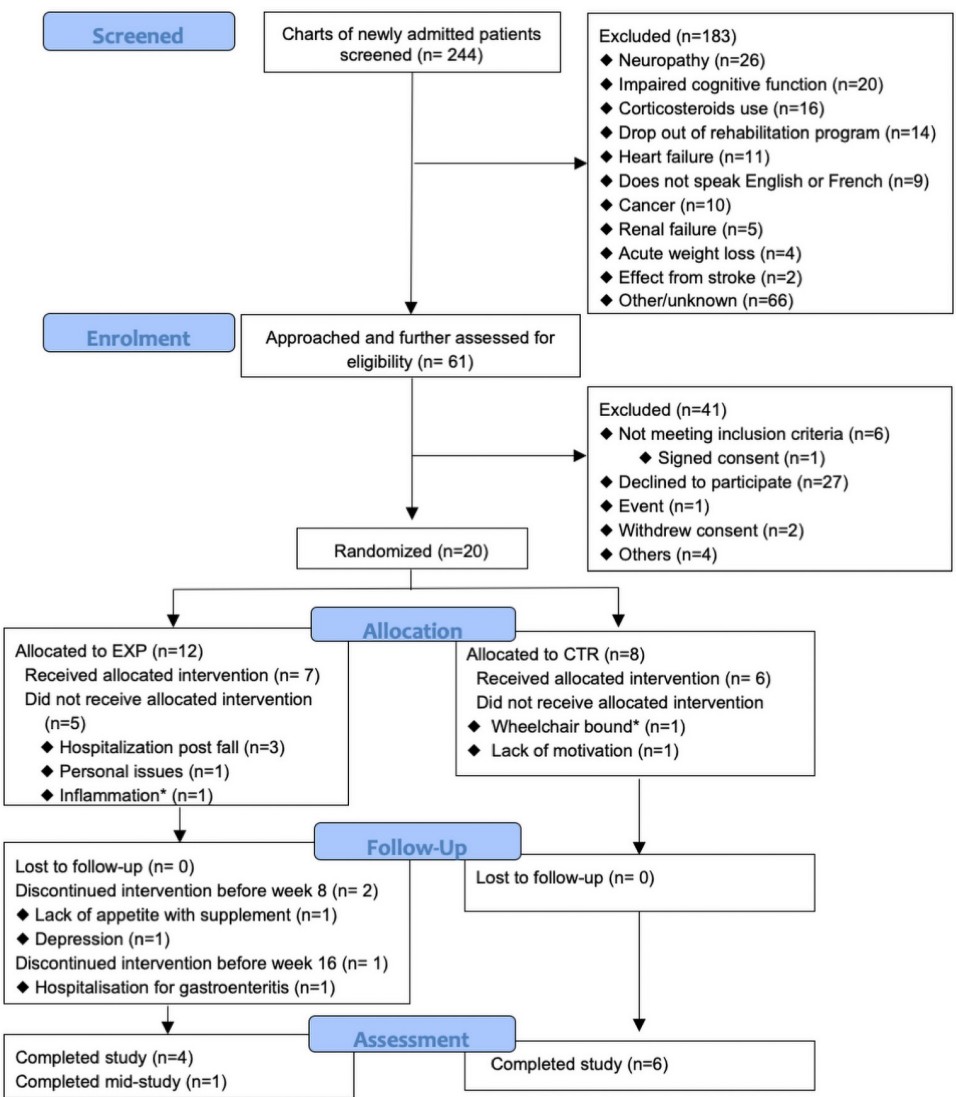

**Fig 1. CONSORT flow diagram of the progress of participants through the study phases, conducted between August 2016 and August 2018.**

significant at week 16 (median: 1.5% of total phospholipid fatty acids, mean: 1.9%, 95%CI 0.35, 2.23) with no change in EPA and DHA proportions in CTR (Fig 2). Total plasma phospholipid fatty acids increased at week 16 in CTR (mean: +130.4 μg/mL, 95%CI: 3.1, 257.6) and tended to decrease in EXP (mean: -180.2 μg/mL, 95%CI: -364.8, 4.5). Serum 25(OH)D did not significantly change within groups (Fig 2).

One participant in EXP reported recurring diarrhea and therefore ceased taking the oil between week 0 and 8 and was later hospitalised for gastroenteritis which confirmed no adverse effect to the supplement. One EXP participant dropped out of the study complaining of low appetite caused by the powder supplement and another reported gastroesophageal reflux at bedtime which seemed to have been related to their eating pattern. One CTR participant developed pneumonia, not related to the supplement and another had high unexplained CRP value at the end of the study.

**Table 1. Baseline descriptive characteristics of participants.**

| Sex | | 5M/15W | |
|---|---|---|---|
| Age, y | | 87 (77, 94) | |
| Education, y | | 13 (2, 23) | |
| Height, m | | 1.57 (1.48, 1.75) | |
| Weight, kg | | 66.1 (49.6, 102.6) | |
| BMI, kg/m$^2$ | | 25.4 (20.6, 36.9) | |
| Walking aid, n (%) | | 17 (85) | |
| Sarcopenic, n (%) | | 5 (26) | |
| Dynapenic, n (%) | | 11 (55) | |
| Frail, n (%) | | 5 (26) | |
| MNA-SF, out of 14 | | 12 (7, 14) | |
| Geriatric depression scale, out of 15 | | 6 (2, 15) | |
| Average daily step count | | 1406 (121, 3344) | |
| **Physical function** | | | |
| TUG, s | | 21.9 (10.1, 53.3) | |
| 6-minute walk test, m | | 218 (68, 415) | |
| Gait speed, m/s | | 0.7 (0.2, 1.4) | |
| Chair stand, number | | 0 (0, 9) | |
| **Cognitive status** | | | |
| MMSE, out of 30 | | 27 (23, 30) | |
| MoCA, out of 30 | | 23.5 (16, 29) | |
| **Clinical markers** | | | |
| 25(OH)D, nmol/L | | 74.0 (44.0, 159.0) | |
| Albumin, g/L | | 40 (32, 48) | |
| Pre-albumin, mg/L | | 221 (135, 339) | |
| IGF-1, ng/mL | | 93.2 (79.8, 144.3) | |
| CRP, mg/L | | 1.4 (0.4, 56.8) | |
| Insulin, pmol/L | | 28.9 (8.8, 102.8) | |
| Glucose, mmol/L | | 5.1 (3.4, 7.1) | |
| | | Men | Women |
| | | $n = 5$ | $n = 15$ |
| Waist circumference (cm) | | 89.0 (83.0, 130.7) | 83.5 (68.0,108.0) |
| **Body composition** | | $n = 4$ | $n = 15$ |
| Total lean mass (kg) | | 45.8 (36.9, 56.1) | 38.0 (28.7, 50.5) |
| Appendicular lean mass (kg) | | 20.0 (15.2, 25.9) | 15.8 (11.8, 24.3) |
| Appendicular lean mass index (kg/m$^2$) | | 7.04 (6.74, 8.43) | 6.37 (4.85, 9.55) |
| **Strength** | | $n = 5$ | $n = 15$ |
| Handgrip (kg) | | 36.0 (22.0, 38.0) | 20.0 (6.0, 27.0) |
| Knee extension (Nm) | | 116.4 (71.8, 151.8) | 59.0 (45.3, 84.2) |

Values are medians (range), $n = 20$. BMI, body mass index; MNA-SF, mini nutritional assessment-short form; TUG, timed-up and go; MMSE, mini mental state examination; MoCA, Montreal cognitive assessment; 25(OH)D, 25 hydroxyvitamin D; IGF-1, insulin-like growth factor 1; CRP, c-reactive protein.

## Physical and body composition assessments

All participants were able to complete the DXA scans, though most found it uncomfortable to lie flat on the scanner bed. Only one participant was not able to perform the TUG test at baseline (9.1%) and all were able to complete 6MWT, gait speed, handgrip and knee extension

**Table 2. Anthropometric measures, step count, and serum clinical markers of participants by group.**

|  | CTR | | | EXP | | |
|---|---|---|---|---|---|---|
|  | **Week 0** | **Week 8** | **Week 16** | **Week 0** | **Week 8** | **Week 16** |
| Sex | 1M/5F | - | - | 1M/4F | - | - |
| Age | 88.5 (77.0, 91.0) | - | - | 85.0 (81.0, 94.0) | - | - |
| BMI, kg/m2 | 28.4 (20.5, 36.9) | 28.3 (20.7, 37.6) | 28.6 (21.3, 38.3) | 22.5 (21.5, 25.4) | 22.9 (21.2, 25.9) | 22.8 (21.5, 25.9) |
| MNA-SF, out of 14 | 12 (9, 14) | - | 13 (11, 14) | 11 (7, 12) | - | 13 (13, 14) |
| Average daily step count | 2133 (121, 3304) | 2088 (229, 3610) | 1522 (496, 2509) | 965 (352, 3344) | 1754 (406, 2309) | 1230 (418, 2174) |
| **Clinical markers** | | | | | | |
| Albumin, g/L | 41.5 (40.0, 44.0) | 41.5 (39.0, 42.0) | 39.0 (32.0, 40.0) | 38.0 (32.0, 48.0) | 39.0 (34.0, 45.0) | 36.5 (35.0, 40.0) |
| Pre-albumin, mg/L | 238 (219, 317) | 262 (200, 305) | 201 (163, 286) | 194 (154, 246) | 238 (132, 246) | 181 (118, 246) |
| IGF-1, ng/mL | 83.9 (65.4, 142.6) | 119.9 (95.4, 144.3) | 81.3 (45.7, 118.8) | 74.3 (31.7, 104.2) | 85.4 (79.8, 91.0) | 91.4 (75.8, 97.3) |
| CRP, mg/L | 1.5 (0.6, 10.1) | 1.3 (0.8, 5.1) | 2.0 (1.2, 11.8) | 2.4 (0.5, 7.5) | 2.0 (0.7, 8.9) | 3.5 (1.0, 62.4) |
| Glucose, mmol/L | 5.3 (4.5, 7.1) | 5.5 (4.8, 7.6) | 5.3 (4.8, 6.9) | 4.6 (3.4, 5.2) | 4.9 (4.4, 5.6) | 5.4 (4.8, 5.7) |

Values are medians (range). BMI, body mass index; MNA-SF, mini nutritional assessment-short form; IGF-1, insulin-like growth factor 1; CRP, c-reactive protein.

strength assessments at all 3 time points. Six (54.5%) participants were unable to execute at least one chair stand.

Table 3 reports data on physical performance, strength and body composition measures and Fig 3 illustrates the changes in these outcomes. Based on the 95% confidence intervals, no changes in TUG, 6MWT, chair stand, leg strength and ALM were observed in either group. In CTR, we observed a slight increase in gait speed (median: +0.19 m/s; mean: +0.12, 95%CI: 0.01, 0.23) and no change in EXP (median: +0.09 m/s; mean: +0.03, 95% CI: -0.16, 0.22). Handgrip strength decreased by 3.0 kg (mean: -2.5 kg, 95%CI: -4.3, -0.6) in EXP and did not change in CTR. While body weight increased in both groups (mean: +1.3–1.4 kg, 95%CI: 0.3, 2.3), total lean mass increased in EXP (median: +663.0 g; mean: +508.3 g, 95%CI: 142.3, 874.2) with no change in CTR.

## Dietary data

Two participants did not complete all 3DFDs because it was deemed too demanding. At baseline, dietary intake in energy, protein, carbohydrate, fat, EPA, DHA and vitamin D were not different between groups (S2 Table). Dietary ALA intake tended to be higher in CTR. When including the supplement, the median daily total protein intake increased by 47.9 g (mean change: 45.6 g, 95%CI: 31.0, 60.3) in EXP at week 16 and reached a median intake of 1.94 g/kg/d (range: 1.55–2.09); total energy intake increased by 520 kcal (mean: +452 g, 95%CI: 240, 664). Median total protein intake of CTR was 1.09 g/kg/d (0.30–1.17) at the end of the trial and energy intake did not change when accounting for the supplement.

At baseline, two participants in EXP and two in CTR reported a daily protein intake below the recommended 1.2 g/kg for older adults [33]. At week 16, all EXP participants reported daily protein intake >1.5 g/kg, and all CTR participants were <1.2 g protein/kg/d.

## Discussion

This 16-week pilot randomized controlled trial of a nutritional supplementation in very old adults with low functional capacity was stopped after 2 years. It was deemed non-feasible because of exceedingly low recruitment rate resulting from a limited number of patients admitted at rehabilitation centers and low eligibility rate. Notwithstanding these issues, the adherence to rigorous study assessments and supplements was good, except for the drop in

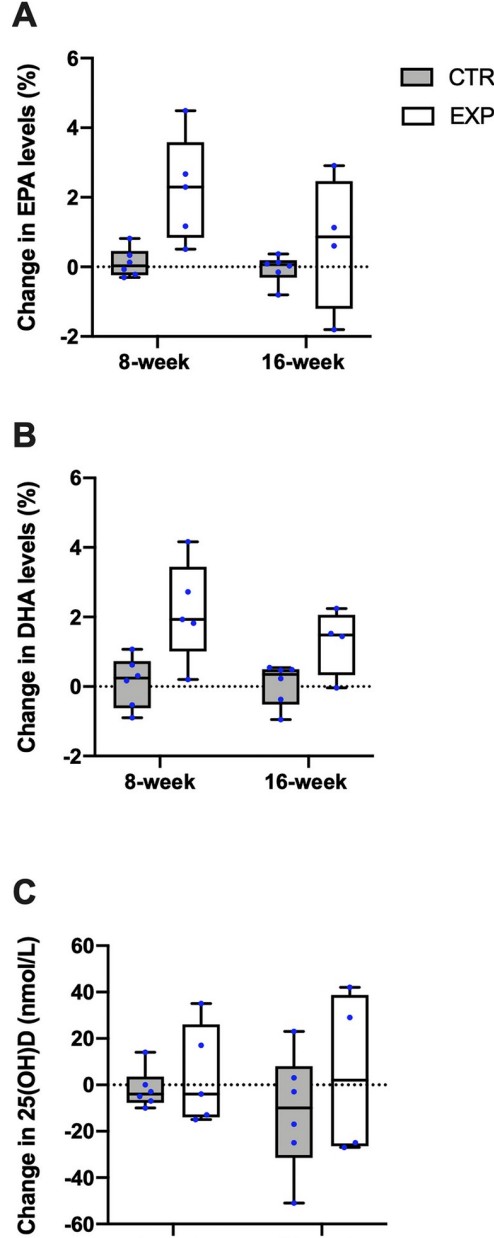

**Fig 2. Boxplots showing changes between baseline and week 8, and week 16 in (A) plasma phospholipid EPA, (B) DHA, (C) serum 25(OH)D by group.** Boxes are interquartile ranges and horizontal lines inside boxes are medians. Whiskers represent the minimal and maximal value and the blue dots represent the participants' individual value. The dotted lines indicate a change of 0.

**Table 3. Physical function and body composition of participants by group.**

| | CTR | | | EXP | | |
|---|---|---|---|---|---|---|
| | **Week 0** | **Week 8** | **Week 16** | **Week 0** | **Week 8** | **Week 16** |
| **Physical function** | | | | | | |
| TUG, s | 22.2 (10.1, 53.3) | 19.5 (10.6, 35.7) | 18.3 (10.1, 44.5) | 19.6 (14.4, 23.2) | 19.8 (12.9, 29.3) | 17.8 (11.4, 21.9) |
| 6-minute walk test, m | 199 (99, 415) | 233 (126, 131) | 201 (91, 423) | 214 (172, 413) | 220 (131, 428) | 263 (228, 364) |
| Gait speed, m/s | 0.6 (0.4, 1.3) | 0.8 (0.5, 1.4) | 0.8 (0.4, 1.5) | 0.7 (0.6, 1.4) | 0.7 (0.6, 1.3) | 0.8 (0.7, 1.2) |
| Chair stand, n | 3 (0, 8) | 3 (0, 8) | 0 (0, 9) | 0 (0, 9) | 0 (0, 11) | 3 (0, 10) |
| **Body composition** | | | | | | |
| Appendicular lean mass, M/W, kg | 18.1 / 15.8 (12.9, 24.3) | - | 16.6/15.8 (13.6, 25.0) | 21.9 / 13.9 (11.8, 15.5) | - | 22.3/13.7 (11.3, 14.9) |
| Appendicular lean mass index, M/W, kg/m2 | 6.74 / 6.37 (5.37, 9.55) | - | 6.19 / 6.53 (5.66, 9.83) | 7.29 / 5.77 (4.85, 6.89) | - | 7.43 / 5.93 (4.63, 6.79) |

Values are medians (range).

fish oil adherence at 16 weeks; the consent rate was adequate, and no major adverse events occurred.

## Primary outcome: Feasibility and potential amendments to study design

Given that nutrition counseling was not an integral part of the rehabilitation programs, the objective of an eventual larger RCT was to test the impact of a multi-nutrient supplement when embedded in such a program on muscle mass, strength and physical performance. The expected rate of recruitment of 3.3 participant per month was noticeably above the observed 95%CI: 0.3, 0.7. The main limitation was the low number of patients admitted to the program reflective of limited resources. After one year of recruitment at one site, a second site was opened but resulted in only a modest increase in recruitment. Access to additional sites would have been necessary to reach the required sample size within a decent timeframe. Consequently, the proposed amendments to the trial include: liberalization of eligibility criteria to increase recruitment; the choice of multi-component endpoints or adjustments of statistical models to improve power; and home visits for provision of supplements, collection of food diaries, activity logs and blood tests, to reduce participant burden and promote retention. The latter strategy has time and financial resource implications to be considered in designing a larger RCT. Exclusion criterion could be more liberal but are nonetheless associated with potential confounding effects on measured outcomes. Hence, the most influential criteria, i.e., renal failure, impaired cognitive function that would prevent an individual from consuming the protein supplement and providing informed consent, respectively, should be retained. Contrastingly, the highly prevalent peripheral neuropathy may be removed as an exclusion criterion as it accurately represents the studied population. Given recruitment challenges and high variability in outcome measures, using a multi-component endpoint as the primary outcome, i.e., combining TUG, 6MWT, gait speed test results, followed by an analysis of specific endpoints controlled for Type I error, or adjustment for prognostic variables should be considered with small sample sizes [34].

High short-term adherence to supplements was confirmed by a marked increase in plasma phospholipid EPA and DHA fatty acid proportion following a 1.9 g fish-oil EPA+DHA supplementation at week 8 (3- and 1.5-fold respectively). This supports effective incorporation into phospholipid, a marker of dietary intake and endogenous fatty acid metabolism. Comparably,

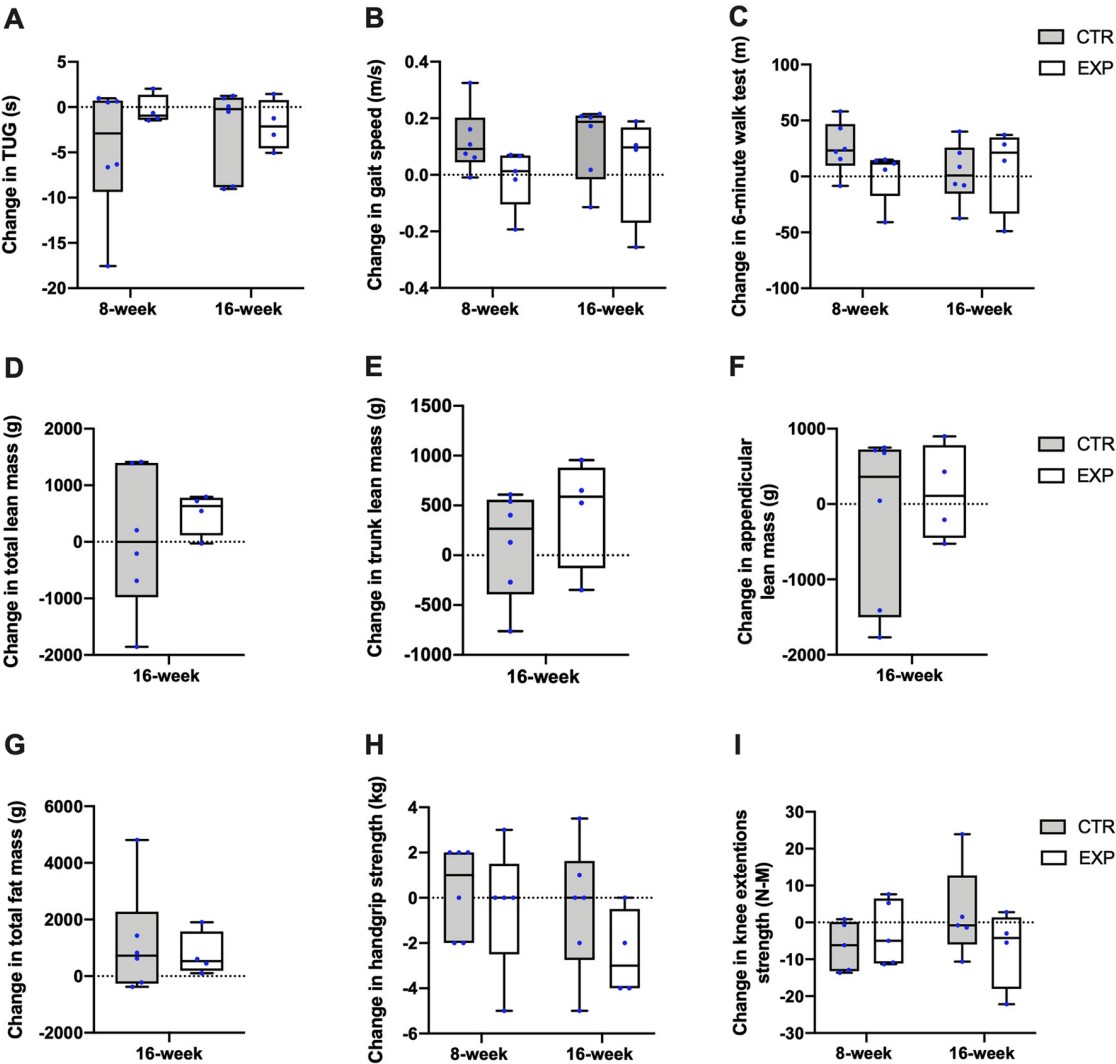

**Fig 3.** Boxplots showing changes between baseline and week 8, and week 16 in (A) Timed up and go (TUG), (B) gait speed, (C) 6-minute walk test, (D) total lean mass, (E) trunk lean mass, (F) appendicular lean mass, (G) total fat mass, (H) handgrip and (I) knee extension strength by group. Boxes are interquartile ranges and horizontal lines inside boxes are medians. Whiskers represent the minimal and maximal value and the blue dots represent the participants' individual value. The dotted lines indicate a change of 0.

Patterson et al. demonstrated a ~85% increase in plasma phospholipid proportion of total EPA+DHA for each additional 1 g EPA+DHA intake in young adults [35].

Serum 25(OH)D levels were adequate at baseline likely due to prevailing vitamin D supplementation in all but one participant. Usual supplements were ceased one week prior to the start of the study which may likely explain the declining trend in circulating levels in the CTR group. The 1500 IU/d dose provided in EXP did not significantly increase serum levels above already high levels (80 nmol/L). Because vitamin D was part of fish oil, the significant increase in EPA and DHA at week 8 and the subsequent decrease at week 16 corroborate self-reported adherence to the oil supplement and reflect a decline in adherence overtime. Self-reported compliance to powder was very good in both groups.

Participants required strong guidance and monitoring from the research team throughout their assessment visit. Assistance to position on the DXA bed was needed and most found

challenging to remain lying flat on their back for the scan. Most participants were not able to complete one chair stand without using their arms, making it impossible to discriminate performance and detect changes overtime. Such difficulties must be considered in study design to avoid a floor effect. The modified 30-s chair stand allowing the use of hands to stand up is a valid and reliable alternative that may be used instead [36]. Handgrip strength and gait speed (+/- assistive device) were feasible and easy to assess in this population and are generally not subject to a floor effect.

Food diaries collected were, for most, incomplete and memory challenges arose when revising diaries with participants. Functional losses in older adults may impede accuracy of dietary assessment, therefore specific approaches tailored to the characteristics of the population may be required [37]. We provided participants with an iPod to capture pictures of their meals as a complement to food diaries. Pictures were used to prompt recall of consumed food items, e.g., cooking method, preparation, and to corroborate reported portion sizes.

## Potential effects on secondary outcomes

Protein, vitamin D and long-chain *n*-3 PUFA have shown potential effects on muscle mass and function separately and were provided together for potential additional or synergistic effects [9]. While studies reported gains in whole-body lean mass (mean: +0.7 kg; duration: 6 weeks; *n* = 38 older men) [12] and ALM (mean: +0.17 kg; *n* = 259 older adults with sarcopenia) [10] from a whey protein and vitamin D-based supplement, another did not (6-month duration, *n* = 60 older adults with sarcopenia) [38]. A mean difference of 508 g in this study is clinically meaningful assuming a total lean mass DXA measurement error of 0.5% [39]. A minimal increment of 180 g had to be detected in our population. Still, it is possible that this result occurred due to shifts in body fluids. Indeed, the larger-scale RCT would be warranted to confirm these results. While knee extension strength did not change in either groups, the median handgrip strength decreased in EXP but not in CTR. This test has shown excellent test-retest reliability [22] and was performed by the same administrator, for all participants throughout the study. This unexpected result may be explained by a decrease in upper body activities during the study, variable motivation during assessment and plausible within-subject variability.

Protein supplementation may have beneficial effects on clinical outcomes including weight, rates of complications and hospitalization, particularly in frail or malnourished/at risk individuals [9, 40]. Thought a recent meta-analysis showed no added benefits from protein supplementation on lean mass and muscle strength [41], a dietary protein intake higher than the RDA (1.6 vs. 0.8 g/kg/d) may preserve lean mass with aging [42]. In the current study, the EXP median protein intake surpassed the recommended 1.2–1.5 g protein/kg/day for older adults [16] without displacing habitual dietary intake, indicating a valuable approach to enhance protein–and vitamin D and n-3 PUFA–intake when needs are high but appetite low, in older adults with normal renal function.

## Characterization of the population

Limited information is available on older adults' n-3 PUFA status. This is the first study to report comprehensive profiling in such a high age group of elderly. Higher plasma, plasma phospholipid and erythrocyte long-chain n-3 PUFA levels were previously reported in older adults compared to younger [43]. Participants in this study appeared to have higher baseline plasma phospholipid EPA (mean: 1.2% EPA) and similar DHA (mean: 2.5%) proportions compared to younger women (0.6% EPA and 3.0% DHA) [44] and older Americans (mean age 74±5 y; 0.5% EPA and 2.9% DHA for middle quintile individuals) [45] from other studies.

Proportions of EPA and DHA, and DHA concentrations, were comparable to those of Canadian patients with advanced cancer close to death specifically (64±10 y; BMI 23.4±4 kg/m$^2$); EPA concentrations were ~25% higher in our population [46]. Interestingly, total plasma phospholipid fatty acids were similarly low in the current study (456±156 ug/mL vs. 442±316 ug/mL in advanced cancer patients [46]) compared to those of healthy individuals. Both diet and metabolism can modify n-3 PUFA proportions. Older adults may have higher levels because they consume more n-3 PUFA sources than younger adults [47] and also due to age-dependent factors altering metabolism such as decreased utilization, greater competition with linoleic acid, higher apparent retroconversion of DHA [48, 49]. The increase in total plasma phospholipid fatty acids was not expected, especially in the CTR group. While adipose tissue loss was previously associated with decreased total plasma phospholipid fatty acids [46], the change observed may be related to the gain in fat mass.

Compared to national data on community-dwelling older adults (aged 65+ y), our participants tended to be older with lower weight and BMI. The prevalence rates of sarcopenia and dynapenia were higher than in the Canadian Longitudinal Study on Aging cohort aged ≥ 75 y (35% vs. 24% and 55% vs. 35%, respectively) [21]. As per TUG, gait speed and chair-stand tests, our population had mobility limitations and was prone to falls [50, 51]. Median TUG of 21.9 s was higher, and gait speed of 0.69 m/s, lower, than means of 9.9–11.0 s at the TUG and 0.92–1.21 m/s gait speed observed in community-dwelling older adult populations [21] including adults >80 y [52]. Performance measured in the present study was similar to that observed in another rehabilitation setting [53]. Lastly, the median 6-min walk test of 218 m is clearly below reference standards of healthy older adults [31] indicating poor endurance and functional capacity.

Conducting an intervention study in this population was challenging. Life events, e.g., acute diseases, death of loved ones, being a spousal caregiver, are stressful especially at an old age and when social support is limited. Isolating the effect of a nutrition intervention may be difficult to achieve in this context. Other limitations include the nature of our study and limited sample size attained which prevents drawing formal conclusions with regards to changes in lean mass, strength and physical performance. Nevertheless, this pilot study has numerous strengths: the robust study design including a nutrient-dense supplement carefully intended to promote muscle health and selected for its taste and ease of preparation; the use of valid and reliable methods to assess plasma phospholipid composition as an objective compliance measure, body composition, strength and physical performance; the characterization of plasma phospholipid n-3 PUFA in this understudied population. Maintaining high adherence to the fish oil supplement up to 16 weeks remains a challenge that could be addressed through other means of delivery, such as part of favorite foods, change of flavors, and more intense follow-up.

In conclusion, due to limited access to potential patients and low eligibility rate, the pilot study was interrupted as deemed non-feasible. However, adherence to the rigorous study assessments and supplements was adequate, except for fish oil at 16 weeks. This RCT pilot study provides the foundation to support the elaboration of a large-scale study needed to investigate potential benefits of a multi-nutrient supplement on lean mass, muscle strength and physical performance. Amendments proposed for a large RCT to be successful in this challenging-to-study population include a multi-center setting and/or larger scale rehabilitation programs, more liberal yet judiciously selected eligibility criteria, allocation of additional time and related resources for close monitoring of participants during and between study visits. The non-amended protocol may be applied to younger or populations with greater autonomy.

## Supporting information

**S1 Table. Proportions of fatty acids in plasma phospholipids of participants by group.** PL, phospholipid; Week 0: CTR $n = 6/6$, EXP $n = 5/5$; Week 8: CTR $n = 6/6$, EXP $n = 5/5$; Week 16: CTR $n = 6/6$; EXP $n = 4/5$.
(DOCX)

**S2 Table. Dietary intake of participants by group.** Week 0: CTR $n = 5/6$, EXP $n = 5/5$; Week 8: CTR $n = 5/6$, EXP $n = 4/5$; Week 16: CTR $n = 5/6$; EXP $n = 3/5$. Daily average dietary intake without accounting for the supplement. M, men; W, women.
(DOCX)

**S3 Table. Nutritional composition of the multi-nutrient supplement and placebo for a one-day provision.** These numbers are applicable for participants weighing 65–75 kg. A factor of 0.8 and 1.2 needs to be applied to the total protein and amino acid profile (except for leucine) to obtain the actual amount consumed for those weighing <65 kg and >75 kg respectively. All amino acids are "L" isomers. *Each participant in the experimental group was given supplements containing both whey protein and leucine powder. Participants received 6 g of leucine powder independently of their weight. Thus, the total amount of leucine per day for participants weighing <65 kg was 10.5 g, and 12.76 g for those weighing >75 kg.
(DOCX)

**S4 Table. CONSORT checklist of information to include when reporting a pilot trial.**
(DOCX)

**S1 File. Study protocol.**
(PDF)

## Acknowledgments

The authors thank the participants of the study and the dedicated health professionals of the Geriatric Day Centres of the McGill University Health Centre and *l'Institut de gériatrie de Montréal*.

## Author Contributions

**Conceptualization:** Anne-Julie Tessier, Stéphanie Chevalier.

**Formal analysis:** Anne-Julie Tessier, Marie Lamarche, Amritpal Bhullar.

**Funding acquisition:** Stéphanie Chevalier.

**Investigation:** Anne-Julie Tessier, Julia Lévy-Ndejuru, Audrey Moyen, Marissa Lawson, Marie Lamarche, Joseé A. Morais, Francis Andriamampionona.

**Resources:** Stéphanie Chevalier.

**Supervision:** Vera C. Mazurak, Stéphanie Chevalier.

**Writing – original draft:** Anne-Julie Tessier.

**Writing – review & editing:** Julia Lévy-Ndejuru, Audrey Moyen, Marissa Lawson, Marie Lamarche, Joseé A. Morais, Amritpal Bhullar, Francis Andriamampionona, Vera C. Mazurak, Stéphanie Chevalier.

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
