## [Decision Letter · Decision Letter 0]

16 Oct 2020

PONE-D-20-25559

A 16-week randomized controlled trial of a fish oil and whey protein-derived supplement to improve physical performance in older adults losing autonomy – a pilot study

PLOS ONE

Dear Dr. Chevalier,

Thank you for submitting your manuscript to PLOS ONE. After careful consideration, we feel that it has merit but does not fully meet PLOS ONE’s publication criteria as it currently stands. Therefore, we invite you to submit a revised version of the manuscript that addresses the points raised during the review process.

Please address each of the reviewers comments. Reviewer 1 had a few major statistical concerns that should be addressed.

We look forward to receiving your revised manuscript.

Kind regards,

Gordon Fisher

Academic Editor

PLOS ONE

Journal Requirements:

2.Thank you for submitting your clinical trial to PLOS ONE and for providing the name of the registry and the registration number. The information in the registry entry suggests that your trial was registered after patient recruitment began. PLOS ONE strongly encourages authors to register all trials before recruiting the first participant in a study.

1) your reasons for your delay in registering this study (after enrolment of participants started);

2) confirmation that all related trials are registered by stating: “The authors confirm that all ongoing and related trials for this drug/intervention are registered”.

3.We note that you have indicated that data from this study are available upon request. PLOS only allows data to be available upon request if there are legal or ethical restrictions on sharing data publicly. For information on unacceptable data access restrictions, please see http://journals.plos.org/plosone/s/data-availability#loc-unacceptable-data-access-restrictions.

Reviewers' comments:

Reviewer's Responses to Questions

**Comments to the Author**

1. Is the manuscript technically sound, and do the data support the conclusions?

Reviewer #1: Partly

Reviewer #2: Yes

2. Has the statistical analysis been performed appropriately and rigorously? 

Reviewer #1: Yes

Reviewer #2: Yes

3. Have the authors made all data underlying the findings in their manuscript fully available?

Reviewer #1: Yes

Reviewer #2: No

4. Is the manuscript presented in an intelligible fashion and written in standard English?

Reviewer #1: Yes

Reviewer #2: Yes

5. Review Comments to the Author

Reviewer #1: This is an interesting article of an unfortunate feasibility study that failed to enroll the desired number of patients and was interrupted due to internal and external factors. The entire endeavor is well-described, and the authors do give some guidance as to full-scale trials that may be relevant in future, if not a direct extension of this one.

The statistical considerations are basically confidence intervals, although the data is longitudinal. But with so few patients, this seems to be appropriate. I will note that the randomization procedure is not given (complete, blocks, etc.), just the software, which is in violation of CONSORT guidelines.

Reviewer #2: The authors report the results of pilot feasibility trial lasing 16wk in older adults at risk of losing their autonomy. The investigators randomized 20 older people to two arms of control treatment or an n-3 fatty acid, vit D, protein combination. 4 subjects completed in the supplement arm and 6 in the control arm. The study was deemed non-feasible.

The concept of autonomy loss in aging is multifactorial. I don’t think this is the most appropriate term to be honest as it fails to classify subjects correctly. Please a more common construct such as sarcopenia, frail/pre-frail, or mobility limited isn’t more appropriate?

The authors report percentages and data to spurious degrees of accuracy. For example, on line 224 they states 26.3% were frail. Surely 0.3% frailty makes no sense and 26% is more sensical. In Table 1, the physical function times reported to the second decimal point are just not possible. Similarly, 6min walk distances to a single decimal place cannot be accurate. Please re-examine the reported data and correct the spurious accuracy of some values.

I think it is reasonable to break Table 2 in tables highlighting physical function and body composition as separate tables. As it stands the table is a rather cumbersome mess of figures and it is difficult to assess what changes and at what time. Note: the figures here are reported to unrealistic levels of accuracy. You cannot measure gait speed to two decimal points nor 6MWT to a single decimal point.

The discussion, at 6.5 pages, is overly long and is very diffuse and unfocussed on the main take home points. There is too much emphasis on the local Montreal influence of why the trial did not work out. That information is fine, but in context of where this paper and the associated data fit it is not overly important. What do people, beyond the investigators themselves, need to be aware of? What changes would you make moving forward to make the trial more feasible? The paper could be made much better with some focussed editing.

Minor

L. 21 “…was interrupted and deemed non-feasible…”

L. 23 Do the authors think the word fitter is appropriate? I might have said with greater autonomy/function/mobility or something similar?

6. PLOS authors have the option to publish the peer review history of their article (what does this mean?). If published, this will include your full peer review and any attached files.

Reviewer #1: No

Reviewer #2: No

---

## [Author Response · Author response to Decision Letter 0]

11 Nov 2020

PONE-D-20-25559 - Response to Editor and Reviewers’s Comments

SC: We thank the reviewers for the thorough revision of our manuscript and insightful comments. We have addressed all points as detailed in this response. (Responses are preceded by SC)

Journal Requirements:

SC: We have revised the manuscript to comply with style requirements and file naming.

2.Thank you for submitting your clinical trial to PLOS ONE and for providing the name of the registry and the registration number. The information in the registry entry suggests that your trial was registered after patient recruitment began. PLOS ONE strongly encourages authors to register all trials before recruiting the first participant in a study.

1) your reasons for your delay in registering this study (after enrolment of participants started);

SC: The following statement has been added (p.5): This trial was registered after participant recruitment began due to the pilot nature of the study (which we did not know required registration at first).

2) confirmation that all related trials are registered by stating: “The authors confirm that all ongoing and related trials for this drug/intervention are registered”.

SC: The following statement has been added (p.5) since no following trial is ongoing : “The authors confirm that there are no ongoing or related trials for this intervention which required further registration.”

3.We note that you have indicated that data from this study are available upon request. PLOS only allows data to be available upon request if there are legal or ethical restrictions on sharing data publicly. For information on unacceptable data access restrictions, please see http://journals.plos.org/plosone/s/data-availability#loc-unacceptable-data-access-restrictions.

SC: As stated at submission, there are ethical restrictions on sharing these data publicly as they derive from a small group of individuals who would be possible to identify with the age, sex and clinical characteristics described, especially men since there were fewer. The REB-approved Informed Consent Form signed by all participants did not include an explicit statement of data sharing, in fact, it precisely specifies that it will be impossible to identify them. Therefore, we will not share individual data to respect confidentiality. 

In an effort to report transparent data and measures of variability as much as possible, we are reporting means + 95% confidence intervals, medians + ranges, and boxplots (median + interquartile range) with individual data points added. 

Comments to the Author

1. Is the manuscript technically sound, and do the data support the conclusions?

Reviewer #1: Partly

Reviewer #2: Yes

2. Has the statistical analysis been performed appropriately and rigorously? 

Reviewer #1: Yes

Reviewer #2: Yes

3. Have the authors made all data underlying the findings in their manuscript fully available?

Reviewer #1: Yes

Reviewer #2: No

Please see answer to point 3 (Journal requirements) above.

4. Is the manuscript presented in an intelligible fashion and written in standard English?

Reviewer #1: Yes

Reviewer #2: Yes

5. Review Comments to the Author

Reviewer #1: This is an interesting article of an unfortunate feasibility study that failed to enroll the desired number of patients and was interrupted due to internal and external factors. The entire endeavor is well-described, and the authors do give some guidance as to full-scale trials that may be relevant in future, if not a direct extension of this one.

The statistical considerations are basically confidence intervals, although the data is longitudinal. But with so few patients, this seems to be appropriate. I will note that the randomization procedure is not given (complete, blocks, etc.), just the software, which is in violation of CONSORT guidelines.

SC: Simple randomization was generated by a staff member not involved in the study using that software. Clarification is added on p.6.

Reviewer #2: The authors report the results of pilot feasibility trial lasing 16wk in older adults at risk of losing their autonomy. The investigators randomized 20 older people to two arms of control treatment or an n-3 fatty acid, vit D, protein combination. 4 subjects completed in the supplement arm and 6 in the control arm. The study was deemed non-feasible.

The concept of autonomy loss in aging is multifactorial. I don’t think this is the most appropriate term to be honest as it fails to classify subjects correctly. Please a more common construct such as sarcopenia, frail/pre-frail, or mobility limited isn’t more appropriate?

SC: We agree that this concept is definitely multifactorial and indeed, the studied population was heterogenous. Consequently, not all participants were frail, or dynapenic, or sarcopenic, or limited in mobility (please refer to Table 1) preventing us from using such terms to characterize the overall studied population. These individuals were referred to the geriatric day hospital rehabilitation program specifically for assessment of “loss of autonomy”, and to provide rehabilitation to prolong independent living and delay institutionalization. For this reason, the term “loss of autonomy” is preferred. 

The authors report percentages and data to spurious degrees of accuracy. For example, on line 224 they states 26.3% were frail. Surely 0.3% frailty makes no sense and 26% is more sensical. In Table 1, the physical function times reported to the second decimal point are just not possible. Similarly, 6min walk distances to a single decimal place cannot be accurate. Please re-examine the reported data and correct the spurious accuracy of some values.

SC: Thank you for this comment; all data have been revised accordingly.

I think it is reasonable to break Table 2 in tables highlighting physical function and body composition as separate tables. As it stands the table is a rather cumbersome mess of figures and it is difficult to assess what changes and at what time. Note: the figures here are reported to unrealistic levels of accuracy. You cannot measure gait speed to two decimal points nor 6MWT to a single decimal point.

SC: This table is now separated in 2 tables, Tables 2 and 3, as suggested.

The discussion, at 6.5 pages, is overly long and is very diffuse and unfocussed on the main take home points. There is too much emphasis on the local Montreal influence of why the trial did not work out. That information is fine, but in context of where this paper and the associated data fit it is not overly important. What do people, beyond the investigators themselves, need to be aware of? What changes would you make moving forward to make the trial more feasible? The paper could be made much better with some focussed editing.

SC: The discussion has been revised thoroughly, refocussed and decreased by one page. Local references are removed. Sub-section titles have been added for better clarity. The main changes suggested to a future trial are summarized in the conclusion. 

Minor

L. 21 “…was interrupted and deemed non-feasible…”

L. 23 Do the authors think the word fitter is appropriate? I might have said with greater autonomy/function/mobility or something similar?

SC: These minor corrections are made.

---

## [Decision Letter · Decision Letter 1]

5 May 2021

PONE-D-20-25559R1

A 16-week randomized controlled trial of a fish oil and whey protein-derived supplement to improve physical performance in older adults losing autonomy – a pilot study

PLOS ONE

Dear Dr. Chevalier,

Thank you for submitting your manuscript to PLOS ONE. After careful consideration, we feel that it has merit but does not fully meet PLOS ONE’s publication criteria as it currently stands. Therefore, we invite you to submit a revised version of the manuscript that addresses the points raised during the review process.

The article has been improved since the first submission but there are still some issues raised by Reviewer #3 which must be addressed. The term “autonomy loss” should be clearly defined in the introduction. It would also help to include criteria used for referral to the out-patient rehabilitation program in the ‘Subjects’ section. The compliance to the oil supplement appears to decreases over time so a more nuanced conclusion about compliance than that it was ‘high’ would be appropriate. Please also address the reviewer’s concerns regarding the content of the abstract

We look forward to receiving your revised manuscript.

Kind regards,

Cameron J. Mitchell, PhD

Academic Editor

PLOS ONE

Journal Requirements:

Additional Editor Comments (if provided):

It would also be more appropriate to title the “Subjects’ section ‘Participants’   

Reviewers' comments:

Reviewer's Responses to Questions

**Comments to the Author**

1. If the authors have adequately addressed your comments raised in a previous round of review and you feel that this manuscript is now acceptable for publication, you may indicate that here to bypass the “Comments to the Author” section, enter your conflict of interest statement in the “Confidential to Editor” section, and submit your "Accept" recommendation.

Reviewer #1: All comments have been addressed

Reviewer #2: All comments have been addressed

Reviewer #3: (No Response)

2. Is the manuscript technically sound, and do the data support the conclusions?

Reviewer #1: (No Response)

Reviewer #2: Yes

Reviewer #3: No

3. Has the statistical analysis been performed appropriately and rigorously? 

Reviewer #1: (No Response)

Reviewer #2: Yes

Reviewer #3: Yes

4. Have the authors made all data underlying the findings in their manuscript fully available?

Reviewer #1: (No Response)

Reviewer #2: Yes

Reviewer #3: No

5. Is the manuscript presented in an intelligible fashion and written in standard English?

Reviewer #1: (No Response)

Reviewer #2: Yes

Reviewer #3: Yes

6. Review Comments to the Author

Reviewer #1: (No Response)

Reviewer #2: (No Response)

Reviewer #3: This article is well written and attempts to address current gaps in the scientific literature related to frailty/sarcopenia/aging and a multi-nutrient supplement containing whey protein, n-3 PUFAs, and vitamin D. It is unfortunate this pilot study failed to recruit the minimal number of participants, resulting in the inability to identify significant findings.

1.) The concept of autonomy loss in aging is multifactorial and unclear. Is there a cut-off for low, moderate, or high autonomy? It seems as if your methodology measures frailty/sarcopenia/dynapenia. In addition, the introduction (line 30, 34, 41, 50) citations refer to a sarcopenia (reference 3, 8, 9, 10) and frailty (reference 4).

A more common identifiable term with reference values would be more useful when applying this work to future research. Assuming all participants do not fall under one category you may identify multiple conditions to monitor. Please choose a more replicable and meaningful term.

2.) The statistical analyses are basically confidence intervals and are displayed as medians and percentiles. I understand that this may be appropriate due to the extremely low sample size. However, the statements in the abstract are misleading. As statistical analyses were not conducted to differentiate significance from non-significance, stating total lean body mass increased and strength decreased in the EXP group is misleading and unclear. In addition, the EXP group had no change in appendicular lean body mass, compared to the CTR group. At minimum, the increase/decrease in LBM, ALM, and strength should be delineated as total change, rather than by “increased and decreased” similar to the other parameters delineated in the abstract.

3.) The abstract does not indicate the number of participants in the CTR versus EXP group. Due to the imbalance and small sample size, this should be stated in the abstract.

4.) I disagree with the conclusion of a high supplement compliance rate. The n-3 PUFAs + Vitamin D oil supplement intake decreased from 86% at week 8 to 55% at week 16 in EXP (line 250). This is a major limitation and should not be described as a success. Decreased compliance may have contributed lack of change in EPA and DHA plasma phospholipids at week 16. If DHA and EPA significantly increased at week 8, but not week 16 is it even relevant? Were these participants losing motivation throughout the intervention, becoming frailer, or fatigued for reasons related and not related to your intervention? How can a gradual loss of compliance be mitigated in the future? These concepts need to be described in the limitation’s section of the manuscript. A high-compliance rate should be removed or modified in the abstract and conclusion.

5.) Did you monitor sunlight exposure? If not, this should be added to the limitation’s sections. This may explain the decrease in the control group at week 16.

6.) Why did you choose the n-3 PUFA and vitamin D doses? You suggest this RCT provides the foundation to support the elaboration of a large-scale intervention. The doses chosen should be adequately justified/referenced in the manuscript.

Minor:

Line 413: add space between younger and reference 39.

Table 1. Baseline descriptive characteristics of participants- Units are missing from parameters including and following “Waist circumference”. Add the appropriate units.

7. PLOS authors have the option to publish the peer review history of their article (what does this mean?). If published, this will include your full peer review and any attached files.

Reviewer #1: No

Reviewer #2: No

Reviewer #3: No

---

## [Author Response · Author response to Decision Letter 1]

3 Jun 2021

RESPONSE TO REVIEWERS - PONE-D-20-25559-R1 

SC: We wish to thank the reviewers once more for the time and efforts invested in reviewing our manuscript and for providing constructive comments and suggestions. We have addressed all questions and comments of reviewer 3 below, and made appropriate modifications to the manuscript. 

Reviewers' comments:

Reviewer's Responses to Questions

Comments to the Author

1. If the authors have adequately addressed your comments raised in a previous round of review and you feel that this manuscript is now acceptable for publication, you may indicate that here to bypass the “Comments to the Author” section, enter your conflict of interest statement in the “Confidential to Editor” section, and submit your "Accept" recommendation.

Reviewer #1: All comments have been addressed

Reviewer #2: All comments have been addressed

Reviewer #3: (No Response)

2. Is the manuscript technically sound, and do the data support the conclusions?

Reviewer #1: (No Response)

Reviewer #2: Yes

Reviewer #3: No

3. Has the statistical analysis been performed appropriately and rigorously? 

Reviewer #1: (No Response)

Reviewer #2: Yes

Reviewer #3: Yes

4. Have the authors made all data underlying the findings in their manuscript fully available?

Reviewer #1: (No Response)

Reviewer #2: Yes

Reviewer #3: No

5. Is the manuscript presented in an intelligible fashion and written in standard English?

Reviewer #1: (No Response)

Reviewer #2: Yes

Reviewer #3: Yes

6. Review Comments to the Author

Reviewer #1: (No Response)

Reviewer #2: (No Response)

Reviewer #3: This article is well written and attempts to address current gaps in the scientific literature related to frailty/sarcopenia/aging and a multi-nutrient supplement containing whey protein, n-3 PUFAs, and vitamin D. It is unfortunate this pilot study failed to recruit the minimal number of participants, resulting in the inability to identify significant findings.

1.) The concept of autonomy loss in aging is multifactorial and unclear. Is there a cut-off for low, moderate, or high autonomy? It seems as if your methodology measures frailty/sarcopenia/dynapenia. In addition, the introduction (line 30, 34, 41, 50) citations refer to a sarcopenia (reference 3, 8, 9, 10) and frailty (reference 4).

A more common identifiable term with reference values would be more useful when applying this work to future research. Assuming all participants do not fall under one category you may identify multiple conditions to monitor. Please choose a more replicable and meaningful term.

The referral of participants to the Geriatric Day Hospital rehabilitation program was based on subjective impression of loss of independence rather than objective measures, though these measures do exist. This explains why we measured and defined frailty, sarcopenia and dynapenia using published cut-offs, precisely to better characterize the heterogenous population under study. The vast majority (85%, all but 3 at baseline) had mobility limitations requiring a walking aid. When examining our data again to find a common feature, it is clear that participants had low functional capacity, as defined by a distance of 400 m or less during the 6-minute walk test, an endurance test (ATS Guidelines). There are no strictly -defined cut-offs but the 400 m threshold is often used to define low functional capacity according to normative data (Casanova et al. Eur Respir J 2011; 37:150-156). Only two participants had distances just above 400 m, at 413 and 415 m. Therefore, we revised our manuscript throughout to qualify our participants as having low functional capacity.

Nonetheless, your valuable comment on the imprecision of the term “loss of autonomy” led us change it 

for “loss of independence” which, from a quick PubMed search, seems to be more commonly used to define the same concept. (We also realized that “loss of autonomy” might come from a literal translation from the French “perte d’autonomie” commonly used in the geriatric jargon). The term may better resonate to readers. We also added more explanation regarding the referral of participants to the program. 

We use the appropriate term loss of independence and tried to clarify its meaning in the introduction (lines 36-40 and 60-64), methods (lines 71-74), results (lines 252-253) and discussion sections. 

2.) The statistical analyses are basically confidence intervals and are displayed as medians and percentiles. I understand that this may be appropriate due to the extremely low sample size. However, the statements in the abstract are misleading. As statistical analyses were not conducted to differentiate significance from non-significance, stating total lean body mass increased and strength decreased in the EXP group is misleading and unclear. In addition, the EXP group had no change in appendicular lean body mass, compared to the CTR group. At minimum, the increase/decrease in LBM, ALM, and strength should be delineated as total change, rather than by “increased and decreased” similar to the other parameters delineated in the abstract.

Indeed, extensive statistical analysis would be inappropriate with such limited sample size. However, stating that LBM increased and that strength decreased in the EXP group is not misleading since the 95% CI of the change from baseline is all positive for LBM (+508.3 g, 95%CI: 142.3, 874.2) and all negative for strength (mean: -2.5 kg, 95%CI: -4.3, -0.6) (Figure 3 and page 17 lines 327-332). We acknowledge that the statement in the abstract may be unclear because it was incomplete: changes were not different between groups (as 95% CI of changes are overlapping). A statistical test (with a p value) is not necessary to perform to visually appreciate these findings in Figure 3. 

Because of word limit in the abstract, that sentence was removed. 

3.) The abstract does not indicate the number of participants in the CTR versus EXP group. Due to the imbalance and small sample size, this should be stated in the abstract.

Number of participants has been added to abstract: 6 CTR, 4 EXP.

4.) I disagree with the conclusion of a high supplement compliance rate. The n-3 PUFAs + Vitamin D oil supplement intake decreased from 86% at week 8 to 55% at week 16 in EXP (line 250). This is a major limitation and should not be described as a success. Decreased compliance may have contributed lack of change in EPA and DHA plasma phospholipids at week 16. If DHA and EPA significantly increased at week 8, but not week 16 is it even relevant? Were these participants losing motivation throughout the intervention, becoming frailer, or fatigued for reasons related and not related to your intervention? How can a gradual loss of compliance be mitigated in the future? These concepts need to be described in the limitation’s section of the manuscript. A high-compliance rate should be removed or modified in the abstract and conclusion.

We agree with this comment and have tempered the “successful” adherence to fish oil supplement at 16 weeks. Because of space limitation in the abstract, we have to pool results of compliance to powder (experimental + placebo) and oil (experimental + placebo) together but a clarification on the decline in adherence to fish oil is now part of the conclusion. DHA was still significantly increased at week 16 but not EPA. We believe that increases at 8 weeks that were not sustained over 16 weeks are relevant; they support acceptance of the fish oil supplement but highlight the need for mitigation strategies to sustain motivation over longer intervention periods. As suggested, a sentence was also added to the paragraph on limitations to propose potential mitigation strategies (p.24 lines 482-484) : “Maintaining high adherence to the fish oil supplement up to 16 weeks remains a challenge that could be addressed through other means of delivery, such as part of favorite foods, change of flavors and more intense follow-up”. 

5.) Did you monitor sunlight exposure? If not, this should be added to the limitation’s sections. This may explain the decrease in the control group at week 16.

Sunlight exposure was not monitored but was likely very limited in both groups given the mobility limitations impeding walking outside, as supported by the low average step count (mean 1406 steps/day). For this reason, we consider the lack of sunlight exposure to have a trivial impact on serum 25(OH)D levels as compared to stopping habitual vitamin D supplements. 

6.) Why did you choose the n-3 PUFA and vitamin D doses? You suggest this RCT provides the foundation to support the elaboration of a large-scale intervention. The doses chosen should be adequately justified/referenced in the manuscript.

The fish oil doses were chosen based on previous studies led by our co-investigator Vera Mazurak having reported maintenance or gain in muscle mass in cancer patients undergoing chemotherapy with ingestion of these doses over an average of 10 weeks (ref 17, Murphy et al. Cancer 2011).

The vitamin D dose of 1500 IU/day was chosen based on an effective dose of 800 IU/d associated with decreased risks of falls (ref 18, Bischoff-Ferrari et al. BMJ 2009), a safe upper tolerable level of 4000 IU/day and is commensurate with prescribed vitamin D supplements to adults of 65 years and older in Québec, of 10,000 IU per week. Please see additions on p.7, lines 139-144.

Minor:

Line 413: add space between younger and reference 39.

Done

Table 1. Baseline descriptive characteristics of participants- Units are missing from parameters including and following “Waist circumference”. Add the appropriate units.

Units have been added.

---

## [Editor Report · Decision Letter 2]

6 Aug 2021

A 16-week randomized controlled trial of a fish oil and whey protein-derived supplement to improve physical performance in older adults losing autonomy – a pilot study

PONE-D-20-25559R2

Dear Dr. Chevalier,

We’re pleased to inform you that your manuscript has been judged scientifically suitable for publication and will be formally accepted for publication once it meets all outstanding technical requirements.

Kind regards,

Cameron J. Mitchell, PhD

Academic Editor

PLOS ONE
---

## [Editor Report · Acceptance letter]

13 Aug 2021

PONE-D-20-25559R2 

A 16-week randomized controlled trial of a fish oil and whey protein-derived supplement to improve physical performance in older adults losing autonomy – a pilot study 

Dear Dr. Chevalier:

I'm pleased to inform you that your manuscript has been deemed suitable for publication in PLOS ONE. Congratulations! Your manuscript is now with our production department. 

Kind regards, 

on behalf of

Dr. Cameron J. Mitchell 

Academic Editor

PLOS ONE